# Talent Identification in Elite Adolescent Ice Hockey Players: The Discriminant Capacity of Fitness Tests, Skating Performance and Psychological Characteristics

**DOI:** 10.3390/sports10040058

**Published:** 2022-04-08

**Authors:** Jean Lemoyne, Jean-François Brunelle, Vincent Huard Pelletier, Julien Glaude-Roy, Gaëtan Martini

**Affiliations:** 1Department of Human Kinetics, Université du Québec à Trois-Rivières, 3351 Boulevard des Forges, Trois-Rivières, QC G9A 5H7, Canada; vincent.huard.pelletier@uqtr.ca (V.H.P.); gaetan.martini@uqtr.ca (G.M.); 2Laboratoire de Recherche sur le Hockey de l’UQTR, 3351 Boulevard des Forges, Trois-Rivières, QC G9A 5H7, Canada; jean-francois.brunelle@uqtr.ca (J.-F.B.); julien.glaude-roy@uqtr.ca (J.G.-R.); 3Service de l’Activité Physique et Sportive, Université du Québec à Trois-Rivières [UQTR], 3351 Boulevard des Forges, Trois-Rivières, QC G9A 5H7, Canada

**Keywords:** performance, athletic development, ice hockey expertise, fitness assessment

## Abstract

Background: The process of talent identification in ice hockey occurs during middle adolescence when players are selected to participate in “off-season” evaluation camps, where coaches observe their fitness levels and status of development. Recently, the Quebec ice hockey federation opted for a holistic approach by evaluating players based on three criteria: (1) fitness, (2) skating abilities and (3) personality traits and psychological assets. This study aimed to analyze the discriminant validity of a multi-dimensional talent identification testing protocol in competitive ice hockey. Method: Data were collected from 160 adolescent hockey players who took part in Team Quebec summer evaluation camps. Off-ice fitness, skating abilities and psychological variables were measured on two consecutive days. Descriptive statistics, group comparisons (gender, positions) and discriminant analyses (selected versus non-selected) were performed. Results: No differences were observed among males in which selected players were similar to non-selected. Results from discriminant analyses also showed no discriminant function for male players. For females, selected players displayed higher fitness, on-ice agility and psychological characteristics. Nine performance markers were significantly discriminant. Conclusions: A holistic evaluation protocol allows for the discrimination of selected and non-selected players in elite ice hockey. Developing more discriminant tests is a promising avenue of research in male ice hockey. Knowing the factors that are associated with team selection in competitive ice hockey allow to focus on the specific attributes to work with young promising players.

## 1. Introduction

Ice hockey involves high-intensity, intermittent actions that require players to perform at superior technical and tactical levels. Accordingly, excelling in ice hockey necessitates a vast repertoire of physical attributes, technical–tactical skills [1] and psychological assets [2]. Regarding the physical aspect, attributes such as aerobic capacity, anaerobic systems, strength, power, speed and agility are required [3]. More specific to ice hockey, Mascaro [4] stipulates that skating speed and agility figure among the most important skills to possess, even if they are used for very short periods [5]. Psychological assets, such as grit [6] and type of personality [7], are also considered when it comes time to adapt to the demands of ice hockey and distinguish the best talents. According to Tarter and colleagues [8], the best way to predict the transition to professional hockey (e.g., National Hockey League) is to consider indicators from different categories such as game performance, skill observation and evaluation, fitness, personality traits and perceptivo-cognitive assets. Indeed, these assumptions suggest that to be efficient, the talent identification process should be designed holistically.

In the last two decades, ice hockey has become increasingly popular internationally and is now played by some 1.5 million registered players from more than 75 different countries. With over 900,000 players under the age of 20 (https://www.iihf.com/en/static/5324/survey-of-players; accessed on 17 December 2021) and hundreds of professional leagues across the world, access to professional hockey is among young hockey players’ long-term objectives. As for most sports, national team selections, world championships and Olympic Games performance and the number of drafted players in the National Hockey League (NHL and others, such as the KHL) were the criteria used by hockey federations to evaluate the quality of their development model. There is a wealth of past research on the fitness status of professional ice hockey players [9,10], which has led to an improved understanding of the game, helped to establish standards and played a role in the evolution of strength and conditioning approaches [11]. However, less is known about the contribution of each component of performance to the definition of sports talent. Can we assume that players who perform better in fitness tests and other assessments, such as on-ice sprint tests and psychometric tests, are prioritized in team selection or professional drafts? Vescovi [12] offers an excellent example of the shortcomings of the NHL combine regarding the associations between players’ fitness and draft status over 3 years. Because certain associations between off-ice and on-ice fitness are plausible [13], further research is needed to verify whether these components can help to determine the best athletes. 

In line with these standards and the complex nature of the sport, the pathway to elite ice hockey is a long-term process that needs to start at a young age [14]. As with most sports development models, the expertise acquisition phase is planned for late puberty, between ages 14 and 16, with the result that many evaluation and development camps become an important step in the talent identification process. The International Ice Hockey Federation (IIHF) development camp (IIHF Development Camp: https://www.iihf.com/en/statichub/4625/development; accessed on 10 December 2021), where the most talented players display their skills before hockey coaches and stakeholders, is a good example. The world’s top hockey countries (e.g., Canada, United States, Finland) also have development camps where players are evaluated at the early stages of sports expertise. Indeed, many countries have specific protocols to conduct periodic evaluation or development camps that serve to guide hockey federations in terms of future national team selections (e.g., under-18, under-20, Olympic Games). Camps such as these are first-rate opportunities to develop performance standards for both off-ice fitness and on-ice skills [15]. In addition, evaluating players at this stage of development is useful to establish the age-group and position-specific standards needed to monitor athletes during their development. Valuable work in this regard has been done in the field of ice hockey. Roczniok and colleagues described and identified variables, such as relative VO_2max_, relative peak power and height, that determined success in the Polish men’s national team selection camp [15]. Recently, Vigh-Larsen et al. [16] conducted a cross-sectional study comparing the best U20 (junior elite) and professional Danish players with a similar sample of U20 Finnish players in terms of anthropometric measures and off-ice and on-ice fitness tests. Interestingly, the most important differences were those observed for anthropometric measures, indicating that professional athletes were taller and heavier, whereas no differences (in fitness and skating) were observed relative to players’ position. This demonstrates that the talent identification process is a complex one [17] and that analyzing the profile of U20 players may have limitations since it very likely starts at earlier stages of development [18,19]. As a result, we believe that older players had already experienced the selection process, thus hindering the real value of testing players based on the multiple factors that determine talent for ice hockey. A recent review by Huard Pelletier and colleagues showed that the transfer from fitness testing to on-ice skating performance is relatively well established [20]. However, few authors have verified the value of fitness testing results in the settings of the talent identification process. Despite the popularity of fitness testing in the athlete development processes, little is known about the relative contribution of each test for athlete selection and/or talent identification. In other words, what factors help to differentiate the best players (or prospects) from the average ones? Bracko [5] offered some insights on the issue by showing that there were significant differences between the skating abilities of elite and non-elite players. However, as Johnston reported [18], the concept of sports talent is multifaceted and should be examined further to better define it from a more global perspective. In this regard, viewing sports talent holistically enables us to identify and combine multiple methods and approaches that are conducive to assessing (and evaluating) its multiple components. Therefore, we define sports talent based on hypotheses from past research to the effect that sports success results from a combination of physical, technical, tactical and psychological assets [8,21]. 

Despite the multiple assessment methods and their usefulness in the talent identification process, less is known about the discriminant capacity of psychological variables. Mustafovic and colleagues [22] suggested that talent identification and selection processes could benefit from the incorporation of more psychological variables to get a better sense of players’ true worth. A good example is rugby, where several psychological variables, including perfectionism, learning strategies, motivation and mental health, are part of the talent identification process [23]. With regard to ice hockey, Gábor [24] offered some interesting insights by testing the discriminant ability of on-/off-ice motor skills and psychological assets to explain success and performance among the best Hungarian players under 18 years old. The measurement of psychological characteristics using validated tools is far superior to the qualitative observation of players that is often used by recruiters or coaches and lacks consistency [25]. Consequently, we believe that a more holistic approach that includes testing protocols that consider the multidimensionality of sports talent is a promising way to improve our understanding of talent identification in ice hockey. The addition of psychological variables measured with validated tools would complete the profile of these athletes, making sure that the best are ultimately retained while allowing stakeholders to better supervise the players and ensure they are mentally healthy.

This study aimed to investigate the athlete evaluation process in Canadian hockey (e.g., Quebec) in the early phase of the talent identification process. Because ice hockey is Canada’s national winter sport, excellence is achieved through the rigorous observation and monitoring of player development. Every four years, each province develops a team composed of players under 15 years old for males and under 16 years old for females to compete at the Canada Games. In other years, Hockey Canada asks the provincial federations to organize development camps regarding the evaluation process for all international events, including the Hlinka Gretzky Cup for men, the Summer Series against USA for women and, of course, the World Junior Championships (https://www.iihf.com/en/tournaments; accessed on 4 January 2022). The purpose of this study is twofold. First, it aimed to describe the fitness, on-ice skating abilities and psychological characteristics of the male and female adolescent players that were pre-selected to take part in the 2021 Équipe Québec development evaluation camp: https://www.hockey.qc.ca/fr/page/excellence/equipe_quebec.html; accessed on 21 July 2021). 

The camp serves as the first phase of team selection for national competitions involving provincial teams. Second, it aimed to verify the discriminant capacity of each attribute tested in the evaluation camps. In summary, this investigation can potentially strengthen our understanding of the talent identification process in the early stages of competitive hockey. Since sports talent is multidimensional [8,23], we hypothesized that some key variables would be discriminant and help to differentiate selected and non-selected players at the end of the camp.

## 2. Materials and Methods

### 2.1. Sample and Procedures

This study was developed in collaboration with researchers and the governing bodies of Hockey Quebec, the province’s ice hockey federation. A total of 199 players between the ages of 14 and 16 years (86 boys: 43%, 14 years old; 113 girls: 57%, 16 years old) were invited to the Team Quebec evaluation camp. The criteria for invitation refer to players’ regular season performances. Indeed, both prospect (e.g., male and female) camps serve as an important talent identification stage to determine those who will represent Quebec in national competitions. A week before each camp, players were informed about the research project during an online information meeting. Those who agreed to participate were asked to sign a consent form (if <16 years old, the parents signed). The project was approved by the ethics board of the researchers’ institution (CER-21-278-07.09). The full protocol included three categories of measures (as described in the following section): (1) off-ice fitness tests (8 tests), (2) on-ice skating abilities (2 tests) and (3) psychological attributes (6 measures). The testing procedure was completed on three separate days (day 1—pre-camp questionnaire, day 2—off-ice fitness tests, day 3—on-ice tests) during one weekend camp.

### 2.2. Measures

Measures were selected based on two criteria. The first was related to Hockey Canada’s standards to ensure that some tests were aligned with those of other provinces. The second was related to the scientific literature on testing and refers to specific components identified as potential determinants of performance (or talent) in ice hockey. For off-ice fitness, ten variables were measured and divided into four categories: (1) anthropometric measures, (2) lower limb power, (3) running and VO_2max_ and (4) upper limb power. For on-ice fitness, two skating tests were used to measure skating speed and agility. Finally, two categories of psychological measures were assessed: personality traits and grit.

#### 2.2.1. Off-Ice Fitness Tests

##### Description of Testing Session

Anthropometric measures are commonly used in player evaluation to assess the body composition of a cohort of ice hockey players [15,16,26,27,28]. To measure height, the athlete stood on the stadiometer platform with their shoes off. Their feet were together and heels were supported on the base of the device, with their arms alongside the body. The participant was instructed to gaze outward, and the measurement was taken following a maximum respiration rate to the nearest 0.5 cm. We assessed body weight simultaneously by instructing the athlete to step on the scale (with shoes off). Results were collected in kilograms (to the nearest 0.1 kg). 

Off-ice fitness testing protocol (see Table 1): Before the testing session, a warm-up was administered and supervised by strength conditioning coaches and trainers who were certified kinesiologists. The warm-up consisted of a 15 min session that was a combination of short runs (8–10 min) and plyometric exercises (5–7 min). Athletes were then invited to perform dynamic stretches for both the upper and lower limbs, as well as a few core activation workouts. The warm-up was performed by both cohorts. 

Muscular power (upper and lower body): Since skating is a core element of performance, several studies have examined lower body power as a predictor of performance [29,30,31,32]. The vertical jump and horizontal jump tests are often cited in studies and appear frequently in the NHL’s combine tests list (NHL Central Scouting Combine fitness results: https://link.nhl.com/centralscouting/public/; accessed on 10 December 2021). Table 1 describes the protocols that were implemented to collect these data. For the vertical jump test, a Vertec (Power Systems, Knoxville, TN, USA) (Power Systems: https://www.power-systems.com/shop/product/vertec, accessed on 10 December 2021) was used to assess this variable. Upper body strength, power and endurance also play a natural role in ice hockey [33]. Indeed, we chose grip strength and pull-ups considering that they are omnipresent in the NHL combine research (NHL Central Scouting Combine fitness results: https://link.nhl.com/centralscouting/public/; accessed on 10 December 2021) [12]. The seated medicine ball throw is now also part of the NHL’s combine and is known as a reliable way to assess upper body muscular power [34]. A Ballistic Ball™ (Assess2Perform, Petoskey, MI, USA) (Move Factor X Ballistic Ball A2P: https://movefactorx.com/, accessed on 10 December 2021) was used in this regard. 

Aerobic capacity, speed and agility: In addition to lower body power, aerobic capacity [35] and running agility [36] are also often assessed in both the scientific literature and the field of testing in ice hockey. We measured running speed by administering the 30 m sprint [37] with a Stalker Pro Radar II (Stalker Sport, Richardson, TX, USA) (Stalker Sport: https://stalker.sport/pro-ii/, accessed on 10 December 2021) combined with Swift Speedlight photocells (Swift Performance, Northbrook, IL, USA) (Swift Performance: https://swiftperformance.com/, accessed on 10 December 2021). For the off-ice agility, we used the Pro-Agility Drill 5-10-5 [38] with a second set of Swift Speedlight photocells.

#### 2.2.2. On-Ice Skating Tests

Only two on-ice tests were chosen for the study. In the interest of efficient time management, this part of the testing protocol was limited to 20 to 24 players on the ice for 50 min of ice time. The 44 m sprint and the Finnish Vierumaki’s ice hockey centre of excellence skating agility test were selected. Four sessions of ice time were needed to screen all the participants, both males and females.

Skating speed and acceleration: The 44 m sprint [39] was measured with a Stalker Pro Radar II (Stalker Sport, Richardson, TX, USA) (Stalker Sport: https://stalker.sport/pro-ii/ (accessed on 10 December 2021)) combined with three pairs of Swift Speedlight photocells (Swift Performance, Northbrook, IL, USA) (Swift Performance: https://swiftperformance.com/ (accessed on 10 December 2021)), one at the start, a second at 6 m to assess the 0–6 m acceleration and a third at 44 m. Figure 1 illustrates the setup of the 44 m sprint test. In the sprint test, the athlete is in the standing position with one foot behind the starting line. After the Swift Gate turns green and beeps, the athlete can initiate a sprint at any time. They skate as quickly as possible to the 6 m gates, then to the 44 m gates, in a straight line. Two attempts are separated by a 3 min break to achieve the best time. The athlete must be informed that they decelerate only after crossing the sprint distance to obtain the lowest possible time.

Skating agility: We used one of Vierumaki’s Ice Hockey Centre of Excellence’s skating tests (https://iihce.fi/suomeksi/Testaaminen/Pohjola-leiritestit/tabid/1150/Default.aspx#/material/872/2401; accessed on 4 January 2022) (illustrated in Figure 2), which measures agility for a wide range of movements in the hockey player’s repertoire [40]: explosive start, braking, short sprints, sharp/tight turns, open pivots and backward skating (Figure 2). Illustrates the four-step design of the test. For the skating agility test, the six cones are placed in a rectangle shape, with three pairs of two cones and each pair separated horizontally by 9 m and vertically by 7 m. Step 1: Athlete stands with their foot placed behind the starting line. After the Swift Gate turns green and beeps, they can initiate the test at any time. The athlete first skates all the way to the other end of the circuit. Athlete brakes and sprints back to the line of the second pair of cones. Step 2: After this second step, the athlete aims to their right, outside the cone to perform two consecutive short turns around the two cones of the third pair. Step 3: After the short turns, the athlete aims for the second pair of cones, where they perform two open pivots (facing the starting line side) around the cones. Step 4: After the open pivots, the athlete aims for the first pair of cones. Turning around the first one, they pivot completely and skate backward to the third pair, performing a slalom with the second pair. After completing one side of backward skating, they move forward to the other side and repeat the pivot for the backward slalom. After the second side is done, the athlete sprints back to the starting line. The best of two trials is recorded.

#### 2.2.3. Psychological Measures: Personality Traits and Grit

Regarding psychological measures, data were collected by means of online questionnaires (Qualtrics software) and participants were given one week before camp to ensure they were in the right frame of mind. Two psychological constructs were measured: personality traits [41,42] and grit [43,44]. Personality traits were assessed with the French version of the Big Five Inventory (BFI-Fr), which is a 44 item Likert-type scale (1 to 5) that has been used for several years in research and clinical settings and is validated in several different languages and populations. It includes conscientiousness, agreeableness, extraversion, neuroticism and openness to experience, which are traits that are significantly associated with sports performance. For example, a high level of extroversion and conscientiousness and a low level of neuroticism would be expected of an elite athlete in a team sport like ice hockey [43]. Grit is a psychological asset that refers to an athlete’s ability to face competition and not give up [44]. We assessed grit with the Short Grit Scale, which is an 8-item Likert-type questionnaire that has shown solid psychometric properties in past research [44].

### 2.3. Statistical Analyses

We calculated descriptive statistics for all data and verified the distributions to identify potential outliers. We also verified the distribution and checked whether assumptions of normality had been violated. The preliminary results indicated no major deviation from normality regarding all off-ice and on-ice measures. Few data were missing because all players took part in the evaluation camp. To verify the contribution of each part of the protocol related to talent identification, as well as the discriminant capacity of each group of components, discriminant analyses were conducted in three consecutive phases. All analyses were performed using SPSS software and based on the recommendations of Tabachnik and Fidell [44]. Discriminant analysis [45] makes it possible to verify which variables are related to group membership (e.g., selected versus non-selected) by assigning a discriminant function to each variable under study. In this regard, group membership was accorded following the evaluation camp, where 44 males (out of 86) and 54 females (out of 107) were retained for the selection process. Those retained were categorized as “selected” (value = 1) and those not retained as “not selected” (value = 0). After the selection process was completed, analyses were conducted on five sets of variables: (1) anthropometric measures (height and weight), (2) off-ice fitness (lower and upper body strength-power), (3) running ability (aerobic capacity, speed and agility), (4) on-ice skating (skating speed and agility) and (5) psychological factors (personality traits and grit). We conducted the analyses separately to prevent the overlapping of factors and verify the discriminant capacity of each category of measures. For each model, we conducted direct discriminant analyses based on our interpretations based on Box’s M and Wilks’ lambda statistics. Finally, we analyzed the discriminant function for each set of predictors and verified the proportion (%) of correct categories regarding the selected players.

## 3. Results

### 3.1. Sample Characteristics

Table 2 presents the sample’s characteristics. The players invited to each selection camp came from two age-group categories. Males were taller than females (F_height_ = 68.03, *p* < 0.001), but no differences were observed regarding body weight. Proportions for player positions were similar across groups owing to the specificity of ice hockey in which a usual roster is mainly composed of 47 as forwards, 31 as defense and 8 as goaltenders. For the male cohort, goaltenders tended to be taller and heavier than defense and forwards (F_height_ = 6.75, *p* = 0.002; F_weight_ = 6.38, *p* = 0.003). No player-position-related differences were observed regarding the anthropometric profile of the female cohort. 

### 3.2. Objective 1: Fitness, On-Ice Abilities and Personality Traits of Elite Adolescent Hockey Players

Table 3 shows the descriptive statistics of each of the testing protocol’s variables. Males displayed higher scores than females on all the off-ice fitness tests (all group comparisons were significant at *p* < 0.01 or 0.001). The same pattern was observed regarding the on-ice skating tests, which showed that boys were faster, had better acceleration and were more agile. Male goaltenders also performed better than female goaltenders on the shuffle tests. However, different patterns were observed when comparing participants’ personality traits. The female cohort displayed higher scores for three traits: extraversion (t_(df)_ = 2.41_(187)_, *p* = 0.017), conscientiousness (t_(df)_ = 2.16_(187)_, *p* = 0.032) and openness (t_(df)_ = 3.55_(187)_, *p* < 0.001). Comparisons between the selected and non-selected players (males and females) were presented in the sections on discriminant analyses. Finally, we compared all measures according to the players’ positions (goaltenders, defense and forwards) and found no significant differences (all *p*-values > 0.05), suggesting that players’ fitness and abilities were similar for males and females. 

### 3.3. Objective 2: Discriminant Capacity for Each Facet of the Talent Identification Protocol

#### 3.3.1. U15 Male Selection

Table 4 illustrates the results of the discriminant analyses for the male cohort. As mentioned, no significant results were found (see Appendix A, Table A1). Indeed, no significant differences were observed, even when the mean scores of the “selected” versus the “non-selected” players were compared, with the exception of the med-ball throw, where differences favored the 44 selected players (F_(df)_ = 2.80_(2)_, *p* < 0.10). No significant discriminant functions were noted in each of the five proposed models. In model 4, none of the variables showed a significant discriminant function, but the selected players had higher scores for grit (F_(df)_ = 3.05_(1,75)_, *p* < 0.10). We retained loadings higher than 0.3 to identify the most discriminating variables, which are identified in Table 5. Each of the five models classified 48 to 60% of selected players. 

#### 3.3.2. U16 Female Selection

As shown in Table 5, we found significant results for the female cohort, where some variables had a discriminant effect on the selection process. Group comparisons (see Appendix B, Table A2) revealed that selected players displayed higher scores in 10 indicators, in which 9 of these 10 variables had a significant discriminant function. Except for anthropometric measures (where we found no significant results), other measures such as off-ice fitness, skating abilities and personality traits revealed a significant discriminant function. With regard to off-ice fitness, aerobic capacity (VO^2^_max_), leg power, speed and agility were identified as discriminant factors. On the ice, skating speed and agility had a significant discriminant function. For psychological measures, agreeability and grit have a significant discriminant function. Each of the five models correctly classified 52–74% of selected players. In summary, models 2,3,4 and 5 displayed significant discriminant functions.

## 4. Discussion

In ice hockey, team selection is a complex process that involves the observation and assessment of multiple variables. According to Tarter [8], the combination of game performance, fitness, on-ice attributes and psycho-cognitive factors is probably the best way to determine a player’s potential. This study sheds some light on talent identification for elite adolescent Canadian hockey players. Given the importance of ice hockey in Canada and this country’s culture of evaluation camps, we believe the present study offers an excellent opportunity to refine our understanding of the factors that distinguish players who are prioritized or categorized as “top prospects” in their field. Additionally, this study offers substantial knowledge about the level of Canadian adolescent elite hockey players, especially because of the multiple determinants that were observed (19 variables) and the population studied, which consisted of male and female hockey players. 

The first part of our study shows that fitness measures have similar patterns when male and female participants are compared. As expected, boys displayed higher scores than girls, which can be explained by the physical maturation and physiological aspects that favor male athletes at this stage [46]. With the exception of male goaltenders’ height, we found no fitness differences related to the players’ positions. This result appears to contradict the findings of previous studies, such as that of Daigle et al. [13], who showed potential position-related differences for certain fitness components. In our opinion, young athletes’ stage of development may explain the lack of position-specific differences regarding players’ fitness. Measurements took global fitness measures into account, which was probably insufficient to detect position-specific skills. These results suggest that introducing “position-specific” skills, such as backward skating (for defense) and skating skills with puck control (for forwards) would be relevant for future research. In terms of differences in personality traits, female players scored significantly higher than male players on extraversion, agreeableness and openness, which is consistent with the scientific literature. However, the score for neuroticism was not significantly higher in the female sample, contrary to what was expected [47,48]. 

The second part of the study was designed to verify the discriminant capacity of each variable tested in the selection process. Results from our analyses offer interesting insights, particularly because the substantial differences they revealed were based on gender. In the male cohort, we found that not all five models were discriminatory. At first glance, this suggested that the current protocol with the current group of players did not contribute definitively to the selection process in the male sample. However, certain measures, including muscular strength, aerobic capacity, skating agility/acceleration and grit, tended to be more strongly associated with the players selected during the process. The picture was different in the female sample, as the fitness tests, running skills and skating performance successfully classified between 62 and 74% of the selected athletes. On a different note, the higher grit score of the selected male players was to be expected since this was noted in several studies. However, a higher conscientiousness score would also have been expected regarding these players, as it is the strongest predictor of success in sports and is generally associated with grit. Interestingly, the protocol’s discriminant capacity was significant in the female cohort. The three personality traits that appeared to distinguish the selected female players were agreeableness, conscientiousness and neuroticism, which is plausible in a team sport context. These three traits are generally associated with success in team sports, as conscientiousness predisposes players to invest more effort, a low level of neuroticism provides greater emotional stability under pressure and agreeableness promotes healthy relationships with teammates [49]. The fact that the training camp contained high-level athletes only very likely reduced the discriminatory power of personality traits; a comparison with recreational athletes would possibly have demonstrated much more obvious results. In general, both male and female participants demonstrated the expected profile of low neuroticism and high conscientiousness and agreeableness [48,49,50].

In our view, the divergent results (related to the discriminant capacity of each protocol) can be explained from a gender perspective using socio-demographics. Even if the initial samples (selected and non-selected) were similar in size, they were not similar regarding the representation of the province’s pool of elite players since Quebec had over 450 male players classified as U15–U18 elite players [50]. The numbers were significantly lower for the female group, where 275 (U15–U18) players were categorized as elite. The development structure suggested that 80 boys represented slightly less than 20% of the highly competitive players as compared with 30% of competitive female players. These numbers indicate that the testing protocol was probably less effective for determining the best prospects in more homogeneous groups, as was the case for the male cohort. Our study has certain practical implications. We believe that two approaches are conducive to the development of improved talent identification protocols for differentiating selected and non-selected athletes. The first is to increase the number of tested players (regarding the male cohort) while considering factors such as birthplace and birth month, which may prove that current tests can effectively identify the best U15 prospects. Teoldo and Cardos discuss these issues with regard to Brazilian soccer [51], which is definitely comparable to Canadian ice hockey in terms of the sport’s popularity. The second approach is to integrate tests that could help to differentiate the most gifted players. In this regard, we think that attributes such as the ability to repeat sprints [35] and skating agility in multiple contexts [52] should be considered in future protocols. We also believe the said protocols should include multiple game performance indicators [13] in view of the association between fitness, on-ice skills and game performance. 

Despite its contribution, this study had limitations. The first was the potential selection bias. To participate in the study, players had to be registered in a league governed by Hockey Quebec. This means that the participants were engaged in a very similar pattern of sports development, possibly leading to an almost identical athletic profile, reducing the discriminant capacity of the protocols, especially among males. The Hockey Quebec development model for males was implemented in 2011, whereas its implementation for females is far more recent (2015). Another limitation of the sample concerns the players’ age groups and physical maturation. Even if further analyses from the current database confirm the physical maturation of most participants, we believe the maturation process is an important factor to consider, especially in sports like ice hockey, where physicality prevails. Additionally, we recognize that certain key talent indicators are missing in the present investigation. 

As Tarter stipulates [8], game observation is crucial to the talent identification process. Despite the challenges involved, establishing associations between fitness and on-ice skills with game performance is a promising avenue for future research. In practical terms, knowing which factors are related with team selection is useful for coaches and stakeholders (e.g., scouts, program directors, strength conditioning coaches) in a way that it allows to identify talent at early stages of hockey expertise. In addition, it seems that these factors might vary according players’ gender, which means that same testing protocols are not assessing talent in the same way. Focusing on more specific ice hockey attributes (e.g., repeated sprints, skills with the puck, anaerobic capacity) for male selections might be an interesting alternative. For female talent selection, our results suggest that the suggested protocol is sufficient to select the best players. Even if game observation was outside the scope of this study, we think that systematic observation and the use of advanced technologies for assessing performance in real settings will be prioritized in subsequent stages of this research. We also believe that the inclusion of perceptual-cognitive markers such as reaction time and decision making could contribute to talent identification [53]. Although some promising work has been done in the field, further developments are needed to establish associations with game performance in real settings.

## 5. Conclusions

In Canadian hockey, talent identification starts in late adolescence and is a crucial stage in the development of sport expertise. Player development in Quebec is monitored in off-season (summer) evaluation camps designed to identify the province’s best prospects for future competitions. Knowledge about the discriminant capacity of each component of talent is crucial for stakeholders (e.g., coaches, scouts, program directors) in talent detection camps. This study refined our understanding of talent identification and revealed that the discriminant capacity of protocols differs based on gender. Interestingly, it showed that multiple components of sport talent help to classify the female players who emerged in the first phase of the selection process. Lower body power, upper body strength, running abilities and on-ice skating efficiency are useful components of the selection process. Inversely, such protocols are less conclusive for male cohorts, suggesting that further research is needed in this area. Future investigations can build on this study by focusing on variables assessed through game observation and other key factors for distinguishing the most promising talents.

## Figures and Tables

**Figure 1 sports-10-00058-f001:**
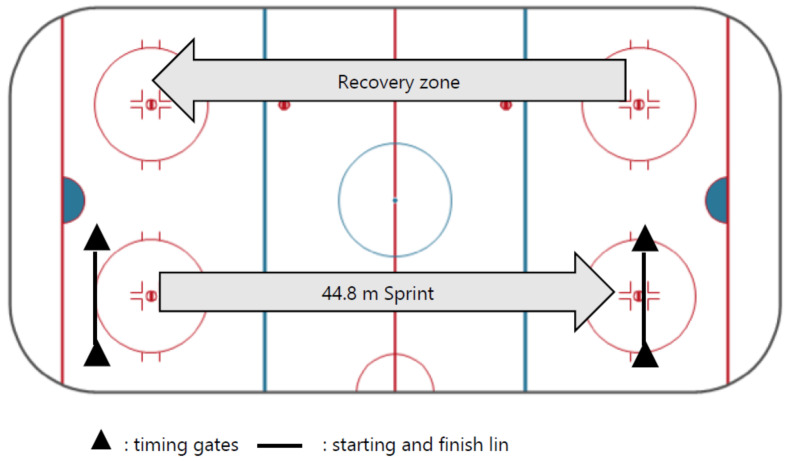
The 44.8 m skating sprint test.

**Figure 2 sports-10-00058-f002:**
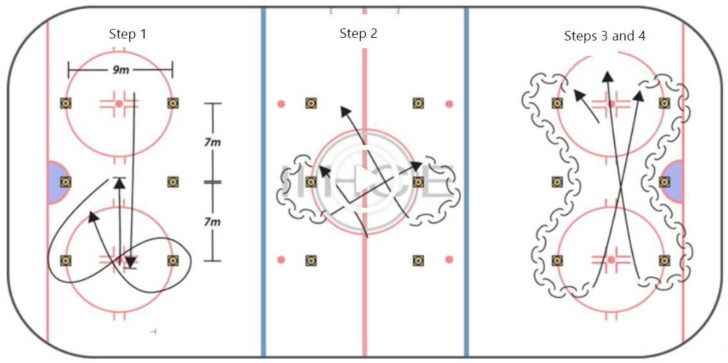
The skating agility test (adapted from the International Ice Hockey Centre of Excellence, Vierumaki, Finland). Illustration shows the three-step procedures, performed on a single circuit. (Video demonstration: https://iihce.fi/suomeksi/Testaaminen/Pohjola-leiritestit/tabid/1150/Default.aspx#/material/872/2401, accessed on 4 January 2022).

**Table 1 sports-10-00058-t001:** Off-ice tests explained.

Variable Test	Measures (Units) [Instrumentation]	Protocol
Lower body power Vertical jump	Height of jump (cm) [Vertec]	Athlete stands under the testing device to measure their maximum height with their arm at maximum flexion. Once the height is recorded, athlete stands at an elbow’s distance from the testing device and performs one pre-trial jump. Athlete bends their legs and pauses for a second before jumping while reaching out as high as possible. The best of 3 attempts is retained.
Broad jump	Length of jump (cm) [Tape on floor]	Athlete positions both feet behind the line. Legs are bent quickly and arms are swung back and forth to initiate the jump. Athlete must land in control, and once stable on their feet, the foot that travels the least distance is measured. The best of 3 attempts is retained.
Aerobic capacity Léger 20 m Shuttle	Maximum oxygen consumption (mL/kg/min) [Tool kit]	The protocol involves the repetition of 20 m shuttle runs to maximal fatigue. Athlete must stand 1 m behind the line before the next beep before changing direction and continue to run They must wait for the signal before starting. The speed increases with each level. Athlete must follow the rhythm of the soundtrack.
Running speed/agility 30 m sprint	Time (s) Acceleration curve (m/s) [Swift timing gates] [Stalker Radar]	Athlete stands with their foot behind the starting line. When the Swift Gate turns green and beeps, athlete can initiate the sprint at any time. They sprint as quickly as possible to the finish line. Athlete makes two attempts separated by a 3 min break to achieve their best time. Athlete must be informed that they should decelerate only after crossing the sprint distance to obtain the best possible time.
5-10-5 agility drill	Time (s) [Swift timing gates]	Athlete places their foot on starting line. They then move as quickly as possible to a line from the end past the photocells and touche the line with their hand. Must move as quickly as possible to the line at the other end. Athlete touches the line with their hand before returning to the center as quickly as possible (best of 2 trials).
Upper body strength Grip strength	Sum of grip test (kg) [Dynamometer]	Dynamometer is set to 0 before each test. Athlete grabs dynamometer in a neutral grip and keeps it close to hips. Exhales heavily and compresses it as hard as possible. Athlete then returns the dynamometer to the evaluator who records the result and resets it at 0 for the next test. Two attempts per hand are allowed and the best result per side is retained.
Vertical pull up	Maximum reps (*n*) [Bar for hanging]	Athlete hangs from the bar with arms straight and performs an arm/back pull to raise the body and lift the chin over the bar. They then descend, fully unlocking the elbows before the next repetition. Count to 2 consecuitve repetitions before stopping the test. Swings and help from the legs are not allowed.
Seated medicine ball-throw	Power (W) [Move Factor Ballistic Ball A2P]	Athlete sits on the ground with back against wall and feet separated by a distance of 60 cm. They hold the ball against their chest at the sides, a little behind the center, with forearms parallel to the ground. They then throw the ball as hard as possible while keeping their back against the wall. The distance of the throw is recorded with the accelerometer in the ball. Three attempts are allowed and the best result is retained.

**Table 2 sports-10-00058-t002:** Sample characteristics (mean (M) and standard deviation (SD)).

		*n*	Age (M ± SD)	Height (m)	Weight (kg)
Males ^+^	Goaltenders	8	13.81 ± 0.40	1.79 ± 5.41 *	71.68 ± 7.54 *
	Defense	31		1.73 ± 7.13	67.35 ± 9.81
	Forwards	47		1.70 ± 6.98	62.09 ± 7.69
	Total	86		1.72 ± 7.37 ^++^	64.88 ± 9.03
Females	Goaltenders	11	14.97 ± 0.92	1.65 ± 3.72	63.72 ± 8.88
	Defense	41		1.65 ± 5.71	64.18 ± 8.73
	Forwards	61		1.64 ± 5.97	61.41 ± 8.48
	Total	113		1.64 ± 5.67	62.64 ± 8.63

* *p* < 0.01. ^+^ Males were taller than females (*p* < 0.001); ^++^ Goaltenders were significantly taller.

**Table 3 sports-10-00058-t003:** Sample characteristics regarding off-ice tests for players who took part in Team Quebec’s summer selection camp.

	Males				Females			
	G (*n* = 8)	D (*n* = 31)	F (*n* = 47)	T (*n* = 86) **	G (*n* = 11)	D (*n* = 41)	F (*n* = 61)	T (*n* = 113)
Off-ice (fitness)								
VO_2_ (mL/kg/min)	46.69 ± 5.05	49.63 ± 4.87	50.35 ± 4.76	49.75 ± 4.88	44.59 ± 4.08	43.86 ± 5.54	44.49 ± 5.74	44.28 ± 5.49
Broad jump (cm)	226.33 ± 14.97	226.33 ± 14.87	227.62 ± 15.25	226.72 ± 14.44	194.10 ± 10.12	193.07 ± 15.00	196.27 ± 15.79	194.88 ± 15.04
Vertical jump (cm)	47.70 ± 5.19	49.04 ± 6.35	48.28 ± 7.11	48.46 ± 6.62	43.64 ± 5.66	42.87 ± 5.82	44.58 ± 6.27	43.87 ± 6.05
Grip strength (kg)	102.50 ± 19.53	99.71 ± 16.46	92.79 ± 16.23	95.83 ± 16.79	71.09 ± 11.88	74.10 ± 10.39	71.78 ± 15.39	72.56 ± 13.40
Chin-ups (*n*)	6.63 ± 4.14	4.19 ± 9.13	9.19 ± 3.22	8.93 ± 3.71	1.36 ± 2.01	2.68 ± 3.11	3.07 ± 2.95	2.76 ± 2.95
Ball throw (W)	165.50 ± 18.88	167.35 ± 19.24	171.85 ± 30.82	169.64 ± 26.08	129.64 ± 17.93	141.27 ± 18.83	134.80 ± 18.39	136.66 ± 18.74
30 m sprint (s)	4.79 ± 0.22	4.75 ± 0.19	4.79 ± 0.21	4.78 ± 0.20	5.19 ± 0.23	5.13 ± 0.23	5.05 ± 0.23	5.09 ± 0.23
Agility 5-10-5 (s)	5.29 ± 0.17	5.36 ± 0.18	5.29 ± 0.19	5.31 ± 0.19	5.71 ± 0.30	5.73 ± 0.27	5.65 ± 0.26	5.69 ± 0.27
On-ice (skating)								
0–6 m accel (s)	n/a	1.31 ± 0.08	1.32 ± 0.07	1.32 ± 0.07	n/a	1.42 ± 0.78	1.43 ± 0.09	1.42 ± 0.09
44 m sprint (s)	n/a	6.25 ± 0.17	6.28 ± 0.25	6.27 ± 0.22	n/a	6.71 ± 0.29	6.77 ± 0.31	6.75 ± 0.30
Agility (s)	n/a	35.54 ± 1.12	35.34 ± 1.40	35.42 ± 1.29	n/a	37.73 ± 1.53	38.27 ± 2.30	38.05 ± 2.04
Shuffles ^δ^ (s)	18.31 ± 1.58 *	n/a	n/a	n/a	19.80 ± 1.19	n/a	n/a	n/a
Recoveries ^δ^ (s)	28.25 ± 2.87 *	n/a	n/a	n/a	31.51 ± 3.24	n/a	n/a	n/a
Psychological variables (5-point scales)								
Extraversion	3.57 ± 0.37	3.33 ± 0.28	3.31 ± 0.60	3.34 ± 0.34	3.49 ± 0.58	3.52 ± 0.34	3.44 ± 0.39	3.47 ± 0.39 ***
Conscientiousness	3.70 ± 0.21	3.66 ± 0.30	3.46 ± 0.28	3.56 ± 0.30	3.69 ± 0.43	3.67 ± 0.34	3.65 ± 0.31	3.66 ± 0.33
Neuroticism	2.62 ± 0.37	2.69 ± 0.41	2.64 ± 0.52	2.66 ± 0.47	2.75 ± 0.60	2.79 ± 0.49	2.69 ± 0.46	2.73 ± 0.48
Agreeableness	2.86 ± 0.24	3.19 ± 0.54	3.10 ± 0.38	3.11 ± 0.44	3.13 ± 0.51	3.15 ± 0.51	3.10 ± 0.50	3.12 ± 0.50 ***
Openness	3.30 ± 0.35	3.26 ± 0.59	3.28 ± 0.45	3.28 ± 0.50	3.55 ± 0.43	3.63 ± 0.46	3.45 ± 0.48	3.53 ± 0.47 ***
Grit	4.13 ± 0.61	3.98 ± 0.45	3.93 ± 0.61	3.97 ± 0.56	4.08 ± 0.61	3.94 ± 0.54	3.76 ± 0.61	3.85 ± 0.60

G: goaltenders; D: defense; F: forwards; T: total. * Boys displayed better scores on goaltenders’ tests: *p* < 0.05. ** Boys displayed better scores on fitness and on-ice tests: *p* < 0.001. *** Girls displayed higher scores regarding psychological measures: *p* < 0.05. ^δ^ Goaltenders’ on-ice tests: in the female group, 9 goaltenders completed the test.

**Table 4 sports-10-00058-t004:** Results from the discriminant analyses for the 44 selected male players (standardized discriminant function coefficients).

Variables	Box’s M	Wilk’s Lambda (χ^2^_(df)_)	Discriminant Function		Classification
Model 1: Anthropometry	1.341 ^ns^	0.963 (13.780_(2)_) ^ns^	Weight	1.116	49%
			Height	−0.157	
Model 2: Off-ice fitness	32.620 **	0.963 (2.926_(5)_) ^ns^	Broad jump	0.463	56%
			Vertical jump	−0.015	
			Med-ball throw	0.954	
			Grip strength	0.404	
			Chin-ups	−0.105	
Model 3: Running ability	3.727 ^ns^	0.994 (0.490_(5)_) ^ns^	VO_2max_	0.950	48%
			30 m sprint	0.080	
			50-10-5 agility	0.124	
Model 4: Skating performance	8.098 ^ns^	0.995 (0.346_(3)_) ^ns^	44 m sprint	0.154	51%
			6 m accel	0.361	
			Agility	0.952	
Model 5: Psychological measures	55.073 **	0.889 (8.457_(6)_) ^ns^	Extraversion	−0.184	60%
			Agreeableness	0.318	
			Conscientiousness	−0.287	
			Neuroticism	−0.036	
			Openness	0.427	
			Grit	0.571 *	

** *p* < 0.01; * *p* < 0.10: Selected players displayed higher grit scores. Bold text: loadings > 0.30 represent the most discriminating variables. ^ns^ non significant.

**Table 5 sports-10-00058-t005:** Results of the discriminant analyses for the 54 selected female players (standardized discriminant function coefficients).

Variables	Box’s M	Wilk’s Lambda (χ^2^_(df)_)	Discriminant Function		Classification
Model 1: Anthropometry	1.743 ^ns^	0.996 (0.464 _(2)_) ^ns^	Weight	**0.892**	52%
			Height	0.203	
Model 2: Off-ice fitness	20.661 ^ns^	0.876 (13.889_(5)_) **	Broad jump	**0.879 ****	62%
			Vertical jump	**0.401**	
			Med-ball throw	0.283	
			Grip strength	0.281	
			Chin ups	**0.753 ****	
Model 3: Running ability	7.797 ^ns^	0.711 (35.946_(5)_) **	VO_2max_	**0.829 ****	74%
			30 m sprint	**0.665 ****	
			50-10-5 agility	**0.859 ****	
Model 4: Skating performance	29.497 **	0.651 (40.161_(3)_) **	44 m sprint	**0.651 ****	73%
			6 m accel	**0.494 ****	
			Agility	**0.975 ****	
Model 5: Psychological measures	60.351 **	0.930 (7.921_(6)_) **	Extraversion	−0.110	61%
			Agreeableness	**0.844 ****	
			Conscientiousness	**0.323**	
			Neuroticism	**0.351**	
			Openness	−0.213	
			Grit	−0.043	

** *p* < 0.001: Selected players displayed higher mean scores for the selected variables. Bold text: loadings > 0.30 represent the most discriminant variables. ^ns^ non significant.

## Data Availability

Data in the actual form was used by researchers and collaborators in the actual project. Researchers agreed to give access to data upon request. Those interested should inform the corresponding author at: jean.lemoyne@uqtr.ca.

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
