# Peer review of "Talent Identification in Elite Adolescent Ice Hockey Players: The Discriminant Capacity of Fitness Tests, Skating Performance and Psychological Characteristics"

_sports, 2022, doi:10.3390/sports10040058_

Round 1
Reviewer 1 Report
An interesting paper, though I feel that it is very niche.
Abstract, lines 21-22. What were the differences observed between positions?
Introduction, lines 76-78. What were the findings from Roczniok and colleagues?
Methods, line 165. Incorrect formatting of VO2max.
Methods, line 180-181. Suggest rephrasing. It reads like a 20 minute warmup lasted 12 minutes.
Methods, line 184. Spelling. Should be "lower".
Methods. To what extent were participants familiarised with testing procedures? If only 2-3 attempts were allowed and the best result taken, this does not account for that fact that there could be a learning effect which takes longer than 2-3 attempts to plateau.
Methods, line 245. Spelling. Should be "his/her".
Table 4. It seems misleading to say selected players when this table has all participants, rather than simply the proportion who were retained from the selection process. I would have though that of greater relevance would be a table comparing the values of those selected vs. non-selected.
Discussion, lines 394-415. This paragraph doesn't really mention the fact that for females, some of the fitness and skating measures were also discriminant. I think you could make more of the fact that the model was far more discriminant for females than males.
Author Response
|
Comments |
Answers |
|
Abstract, lines 21-22. What were the differences observed between positions? |
Thank you for your comment. There was no significant difference between positions; we made the correction. |
|
Introduction, lines 76-78. What were the findings from Roczniok and colleagues? |
We added 3 indicators from Roczniok’ article. |
|
Methods, line 165. Incorrect formatting of VO2max. |
We made the necessary changes in the manuscript. |
|
Methods, line 180-181. Suggest rephrasing. It reads like a 20 minute warmup lasted 12 minutes. |
We explained it more clearly in the text. |
|
Methods, line 184. Spelling. Should be "lower". |
We made the necessary change, thank you for pointing it to us. |
|
Methods. To what extent were participants familiarised with testing procedures? If only 2-3 attempts were allowed and the best result taken, this does not account for that fact that there could be a learning effect which takes longer than 2-3 attempts to plateau. |
The majority of the tests used at the selection camp are fairly common and familiar to most players, because this protocol is suggested by the federation. In addition, given the busy schedule of the players during the camp, the number of attempts per tests had to be limited so as not to exceed the time allotted by the federation officials. We also think that the 2-3 attempts for each test represent fairly well what is usually done by professionals on the field. |
|
Methods, line 245. Spelling. Should be "his/her". |
Corrected. |
|
Table 4. It seems misleading to say selected players when this table has all participants, rather than simply the proportion who were retained from the selection process. I would have though that of greater relevance would be a table comparing the values of those selected vs. non-selected. |
We removed the ''selected'' that was in the first row of the table, it was an error. We believe that a comparative table is not relevant given the small number of significant differences and the fact that they are already found in the discriminant analysis. |
|
Discussion, lines 394-415. This paragraph doesn't really mention the fact that for females, some of the fitness and skating measures were also discriminant. I think you could make more of the fact that the model was far more discriminant for females than males. |
Thank you for the comment. We have added a sentence that highlights this fact, and the possible reasons are given 2 paragraphs later. |

Reviewer 2 Report
The work is aimed at understanding the process of talent identification in U15 ice-hockey and reveals the discriminant capacity of some components to classify the female and male players. I find your work well structured and presented, I congratulate you for the neat and rigorous research design.
Some corrections and clarifications are need, my comments and suggestions are listed point by point in the attached file.

Author Response
|
Comments |
Answers |
|
Line 21: you should specify which variables were statistically different |
Thank you for your comment. There was no significant difference between positions, so we made the correction. |
|
Line 29: which technical and tactical levels is? Low, medium, high level? |
Thank you for pointing that out, we meant ‘’high’’. |
|
Lines 144-147: “Players from two different age groups were recruited and invited to summer evaluation camps. A total of 199 players between the ages of 14 and 16 (86 boys: 43%; 113 girls: 57%) were invited to the Team Quebec evaluation camp.” The two paragraphs should be merged avoiding repetitions, also age of the two groups should be specified. |
We decided to remove one sentence and add the age of the participants in the brackets. |
|
Lines 148-149: it appears that there are 2 camps, is this linked to the 2 groups of subjects? |
Yes, there were a male and a female camp, we have now specified it in the text. |
|
Lines 148-150: “Indeed, both prospect camps serve as an important talent identification 148 stage to determine those who will represent Quebec in national competitions. In the case 149 under study, the camp was designed to identify players to compete at the national level.” The two paragraphs should be merged avoiding repetitions. |
We have removed the redundant sentence, thank you for pointing it out. |
|
Line 165 and along the text: check that VO2max is correctly written with the capital O and the number 2 subscript. |
Thank you for pointing that up, we made the required corrections in the text. |
|
Lines 180-181: I guess some units of time are not incorrect, 20-minute of running cannot fit into a 12-minute warm-up. |
We tried to explain it more clearly in the text. |
|
Table 4: check that all data are reported with a full point as decimal separator. And grip strength of females F group value of 3.07 doesn't seem to be correct. |
We made the correction. |

Reviewer 3 Report
This paper examines the discriminant validity of a multi-dimensional talent identification testing protocol in competitive ice hockey. This is a useful and interesting paper about a holistic evaluation protocol to allow the discrimination of selected and non-selected players in elite ice hockey. However, in my opinion, authors must provide some more details to support their observations and findings.
The statistical analysis do not include an inferential analysis between selected and non-selected players, so we cannot see the differences between groups. In the same way, the effect size of the differences has not been included in the manuscript. The statistical analysis is non-existent as to how the data were analyzed in respect to the hypotheses to be tested. What was the test retest reliability e.g., ICCRs SEM and what are your confidence intervals.
The tables must be self-explanatory. The authors have to specify the meaning of ns in Table 6.
Authors should give readers a "take-home" message and you should improve the practical application in this field. Make sure the last sentence in the abstract and the Practical Applications section of the paper make it clear how your findings can be used by the practitioner in the field.